# Prevalence and Risk Factors for Eating Disorders in Adolescent Female Basketball Players

**DOI:** 10.3390/nu17223547

**Published:** 2025-11-13

**Authors:** Karol Isabel Valenzuela-Farias, Juan Carlos de la Cruz-Márquez, José Alejandro Ávila-Cabreja, María Belén Cueto-Martín, Jesús Siquier-Coll

**Affiliations:** 1Department of Physical Education and Sports, Faculty of Sport Sciences, University of Granada, 18071 Granada, Spain; karol18.vf@gmail.com (K.I.V.-F.); belencueto@ugr.es (M.B.C.-M.); 2Foundation for Biosanitary Research in Eastern Andalusia (FIBAO), 18012 Granada, Spain; javila@fibao.es; 3Department of Communication and Education, University Loyola, Andalucía, 41704 Dos Hermanas, Spain; jsiquier.research@gmail.com

**Keywords:** eating behavior, basketball, women, women’s health, adolescent, team sports, physical exercise, body dissatisfaction, mental health

## Abstract

**Background:** Eating disorders (EDs) adversely affect health and athletic performance, yet their prevalence in adolescent female team athletes is understudied. Objectives: This exploratory pilot study assessed ED presence and associated factors in female basketball players aged 10–18 years from a high-level club in Granada, Spain. **Methods:** This cross-sectional pilot study examined 36 participants (mean age 13.3 ± 2.2 years) who completed the Spanish Eating Disorder Examination Questionnaire (S-EDE-Q). Body composition (Tanita TBF-300), sociodemographic characteristics, and sport-related factors were recorded. Associations between these covariates and S-EDE-Q global scores (GS) were examined via linear regression. **Results:** A higher Global Score (GS), indicating increased ED risk, correlated significantly with several factors: Lower paternal education (primary vs. university: *p* < 0.005) and lower socioeconomic level. Older age (AME = 0.30; *p* < 0.005) and greater muscle mass (*p* < 0.001). National competition level (*p* < 0.001), being a player in the pivot position (mean GS = 1.62), and presence of sport-related medical history (mean GS = 1.07). No associations were found with sport experience or weekly training hours. **Conclusions:** Although the overall risk of EDs is low, the association with specific factors (sociodemographic, anthropometric, and sports-related) allows vulnerable players to be identified. Longitudinal and larger-scale research is needed to confirm these initial findings and design targeted and early prevention strategies.

## 1. Introduction

Eating disorders (ED) are intricate psychiatric conditions that significantly influence eating behaviors and body image perception [1]. They have profound effects on the physical and psychological health and performance of athletes [2]. According to the Diagnostic and Statistical Manual of Mental Disorders (DSM-5), eating and eating behavior disorders (EBD) are characterized by persistent disturbances in eating or related behaviors, leading to impaired food intake or absorption and causing significant physical health or psychosocial functioning impairments [3]. The DSM-5 Diagnostic Criteria Reference Guide [4] outlines specific criteria for clinical identification of these disorders, which include (i) Anorexia Nervosa (AN), characterized by restricted energy intake leading to significantly low body weight, an intense fear of gaining weight, and a distorted perception of body weight or shape; (ii) Bulimia Nervosa (BN), defined by recurrent episodes of binge eating followed by inappropriate compensatory behaviors, such as self-induced vomiting or excessive laxative use, and (iii) Binge Eating Disorder (BED) involves recurrent episodes of binge eating without subsequent compensatory behaviors. These conditions, first described by Richard Morton in the 17th and 18th centuries and later defined by William Withey Gull, remain relevant today, particularly in a society where appearance and physical image are often equated with success [5].

In the sporting context, Marí-Sanchis et al. (2022) highlight that the prevalence of ED is significantly higher among athletes compared to the sedentary population [2]. The rise of social and media pressures to conform to specific aesthetic ideals exacerbates the risk of ED, particularly among vulnerable groups such as adolescents and female athletes, necessitating a specialized approach to their study and treatment [4,6,7,8].

Prevalence rates of ED show remarkable global variability. According to 2016 European studies, the prevalence of Anorexia Nervosa (AN) ranged from <1% to 4%, Bulimia Nervosa (BN) from <1% to 2%, and Binge Eating Disorder (BED) from 1% to 4% [9,10]. By 2018, the global prevalence of ED was estimated at 13%, with a significant increase in adolescents and young women [8], and a reported prevalence in Europe of 2.2% (range 0.2–13.1%) [11,12]. This global concern is amplified in the context of competitive athletics. In sports, female athletes, especially in disciplines that value thinness or include weight categories, face an elevated risk. The combination of maintaining low body weight and the physical and psychological demands of intensive training can lead to risky eating behaviors [13]. While aesthetic and endurance sports such as gymnastics and figure skating are most frequently associated with high rates of ED, team sports—such as basketball—also carry significant risks [14,15,16]. Despite being a team sport that does not prioritize extreme thinness, a specific study with female basketball players revealed that 11% had inappropriate eating attitudes, in contrast to 15% of non-athletes. Although this difference did not reach statistical significance, this finding is crucial, as it underscores that female basketball players remain vulnerable to the development of ED due to pressures to maintain optimal performance and the influence of coaches and teammates [7].

The Spanish Public Health Strategy 2022 states that ED affects approximately 4–6% of females aged 12–21 years, compared with 0.3% of males [11]. In Spain, early studies indicated a higher prevalence of ED in females (3.4% to 6.4%) compared with males (0.27% to 1.7%) [12]. Recent studies by Álvarez-Malé et al. [9] and Rojo-Moreno et al. [13] in 2015 found similar results, with a prevalence of AN and BN below 1% (ranging from 0.19% to 0.57%) and of unspecified ED above 3%, with an incidence of new cases of 2.7% among adolescents. A study conducted in Gran Canaria (Spain) reported an ED risk of 27.42% and an overall prevalence of 4.11% among adolescents aged 12–20 years [9].

EDs among adolescents and athletes encompass a broad range of serious physical and psychological health issues that can significantly diminish both quality of life and athletic performance [12]. A longitudinal study indicated that athletes exhibiting EBD face a markedly higher risk of injury, suggesting that these behaviors detrimentally impact both mental health [17] and physical performance [1,7]. These disorders may lead to severe medical complications, including osteoporosis, cardiovascular issues, gastrointestinal problems, and menstrual irregularities, collectively referred to as the female athlete triad. The prevalence of amenorrhea in female athletes participating in sports such as long-distance running, ballet, or figure skating ranges from 25% to 70%, in stark contrast to approximately 5% in the general population [18]. The interplay of insufficient caloric intake, excessive physical activity, and low body fat (BF) can disrupt the normal menstrual cycle, adversely affecting bone health and increasing fracture risk [1]. These health challenges are often intensified by the adoption of harmful weight control practices, including the misuse of diuretics and laxatives, prolonged fasting, and excessive exercise [12,17].

The psychological ramifications of ED are equally significant. Athletes with these disorders frequently experience low self-esteem, distorted body image, and personality traits such as perfectionism [2]. Additionally, these conditions are linked to a higher prevalence of depression, anxiety, and an increased risk of suicidal thoughts.

In addition, ED can lead to decreased energy availability, muscle weakness, concentration and coordination problems, all of which negatively affect athletic performance. In a team sport such as basketball, where physical and mental agility are crucial, ED can hinder individual and team success. On the other hand, ED is often accompanied by anxiety, depression, and low self-esteem, which can further affect the performance and quality of life of female athletes. The pressure to excel in sport can exacerbate these psychological problems [1,2,17,18].

In recent years, several socioeconomic factors have been identified that can act as lenses through which individual vulnerabilities and environmental pressures are filtered and exacerbated (or mitigated) [19]. The relevance of residential or cultural origin, as well as socioeconomic status, lies in the fact that these can influence exposure to risk factors, social norms, and access to healthcare, which can determine exposure to restrictive beauty ideals, aesthetic pressures, or access to transcendental health information and resources in the presence of ED [19].

Our study addresses gender-specific vulnerabilities in female athletes, where body image and weight control are influenced by societal ideals and sport-specific demands [15,16,17,18]. Although research often targets aesthetic or endurance sports, there is a notable lack of data on team sports like basketball. This gap is crucial because the high physical demands of basketball can still generate concerns about weight and body image, particularly in national leagues.

We focused on female basketball players aged 10–18 years, a critical developmental window highly susceptible to ED, where early detection is vital.

The aim of this exploratory pilot study was to assess the presence of ED symptoms in female basketball players aged 10–18 years from a high-level club in Granada, Spain, and to explore their association with relevant individual and contextual factors. Specifically, we examined the relationship between the Global Score of the S-EDE-Q and body composition, eating behaviors, sociodemographic characteristics, competition level, and medical history.

Specifically, we posited the following hypotheses regarding the Global Score (GS) of the S-EDE-Q: A measurable percentage of female adolescent basketball players, despite participating in a non-leanness sport, will show clinically significant indicators of ED risk. Socio-demographic Factors: Higher ED risk (higher GS) will be associated with lower socioeconomic status and lower parental educational attainment. Higher ED risk (higher GS) will be associated with older age and specific anthropometric measures (e.g., higher BMI z-score and greater muscle mass). Higher ED risk (higher GS) will be found in athletes competing at higher competition levels (e.g., national) and in those with a history of sport-related medical issues (e.g., injuries).

## 2. Materials and Methods

### 2.1. Participants

Sample and Recruitment: The sample comprised 36 female basketball players aged 10 to 18 years from the G + B (Granada plus Basketball) sports club, covering all relevant age categories. Sampling Strategy and Size Justification: Participants were recruited via convenience sampling, based on the voluntary participation of eligible players from the club.

This approach was selected due to practical considerations, including accessibility constraints and the exploratory, pilot nature of the study within a localized athletic context. Prior to data collection, the minimum sample size was obtained by estimating the minimum detectable effect. Based on prior literature, the analysis suggested that a minimum of 35 participants would be adequate to achieve the study objectives, setting the desired statistical power at 95% and an alpha level at 0.05. Inclusion and Exclusion Criteria: Inclusion criteria were as follows: Female basketball players aged 10 to 18 years, active club participation (regularly attending scheduled training sessions), signed informed consent from parents/legal guardians, and participants. Exclusion criteria were as follows: Presence of any disability preventing the application of the evaluation instrument, and repeated absences from scheduled training sessions

### 2.2. Study Design

A descriptive, observational, cross-sectional design was used for this pilot and exploratory study. The objective of this study was to evaluate the presence of ED within a defined population of a high-level women’s basketball team from Granada, Spain, at a specific time point. This team comprises the best players from each category in its area, as it is registered in the highest-level national competitions. The study was conducted in the city of Granada (Spain) over a duration of approximately seven months (8 January and 31 July 2024), without any intervention in the phenomenon being investigated.

### 2.3. Data Collection, Questionnaires, and Procedures

#### 2.3.1. Eating Disorders

To evaluate behaviors, attitudes, and feelings related to eating and body image, the Spanish Eating Disorder Examination Questionnaire (S-EDE-Q) was administered under supervision. This shortened, self-report tool assesses the specific psychopathology of ED, serving as the validated Spanish version of the gold-standard Eating Disorder Examination (EDE) clinical interview [19]. The S-EDE-Q comprises 38 items focused on the preceding three months and provides a measure of overall ED severity through a Global Score (GS), which is the mean of four primary subscales: Restraint (R), Eating Concern (EC), Shape Concern (SC), and Weight Concern (WC). A higher overall score indicates greater severity of ED psychopathology. Consistent with validation studies [20], a score of 4.0 or higher on the individual subscales is utilized as the cut-off for clinically significant psychopathology. The instrument is a valuable screening tool for symptom assessment and risk identification in Spanish-speaking populations [19]. Note: Specific behavioral items within the S-EDE-Q were also used to assess dieting behaviors, binge eating episodes, compensatory behaviors (e.g., self-induced vomiting, laxative/diuretic abuse), and excessive exercise.

Data collection was carried out using paper forms, which were anonymized by assigning unique codes to each participant.

#### 2.3.2. Anthropometric Measurements

Measurements were performed in the Faculty of Sports Sciences laboratory in the presence of the female principal investigator. Measurements were taken at the same time of day, between 8:00 and 8:30 a.m., before eating and after the first urination of the day while at rest.

Anthropometric measurements were obtained using a Tanita TBF-300 body analyzer (Tanita Corp, Arlington Heights, IL USA) and a portable measuring rod with a measuring base from 15 to 210 cm.

All anthropometric measurements were performed in duplicate by a member of the team (JCCM; ISAK level 2).

Height was measured using a calibrated vertical stadiometer (accurate to 0.1 cm. Participants were measured barefoot, in lightweight clothing, and positioned in the Frankfort plane. Two measurements were taken, and the average of the two closest values was recorded. Body weight was measured using a calibrated Tanita scale (accurate to 0.1 kg to 0.05 kg following the manufacturer’s instructions. Players stood on the scale with minimal clothing and without shoes. Two measurements were taken, and the average of the two closest values was recorded. The BMI z-score was calculated using the measured height and weight, referencing the WHO growth charts by age [21].

Sociodemographic Factors: Age (10–18 years), gender (females), parents’ education (primary school, high school, university degree, and postgraduate), parents’ occupation (company director, qualified staff, and non-qualified staff), place of residence (Beiro, Atarfe, Zaidín, Churriana, North Zone, Chana, Genil, Downtown) and income level (lower-middle, middle, Upper-middle, High) [22] were the socio-demographic factors [23] and medical history (sports-related or non-sports-related injuries: not specified, non-sport related, sports-related),

Sport-related variables: Covariates related to sports practice were also collected: Playing position (point guard, center, shooting guard, small forward, power forward, not defined); competition level (local, regional, national), and weekly basketball dedication (hours/week) (4, 8, 12, 16 h/w).

Dependent Variable: ED Symptom Severity as Assessed by the S-EDE-Q [3,4].

Materials used: Portable measuring rod, Tanita TBF-300 body composition analyser, Microsoft Word and Excel spreadsheet suite Office 2024 and editing software, and R statistical software version 4.4.0.

### 2.4. Statistical Analysis

Given the pilot nature of this work, we intentionally examined multiple potential covariates to inform the design of future hypothesis-driven studies. The variables considered in this study were as follows: Age at the time of completing the questionnaires, Anthropometric: Weight (kg), height (m), body mass index z-score, body bioimpedance, where the percentage of body fat (%BF) and muscle mass (MM) (Kg) were measured, Sociodemographic: Place of residence, parental education, socioeconomic status and economic data by residence [21]. Related to sport: Position played (point guard, shooting guard, small forward, power forward, center), level of competition (local, regional, national, international), sporting experience (years playing sport), hours spent training per week, medical and psychological history (injuries, illnesses, previous ED, among others).

Descriptive Statistics: Qualitative variables were characterized using absolute and relative frequencies. Quantitative variables were summarized using the mean and standard deviation (for normally distributed data) or the median and interquartile ranges (for non-normally distributed data). The Shapiro–Wilk test was used to assess the normality of quantitative variables.

Regression and Association Models: To explore the relationships between covariates and the Global Score (GS) of the S-EDE-Q, we employed linear regression models.

For quantitative covariates (e.g., age, anthropometric measures), Average Marginal Effects (AMEs) were calculated using partial derivatives to evaluate their impact on the GS.

For the relationship between the GS and mean gross income (€), a linear mixed regression model was utilized, treating the natural logarithm of income as a fixed effect and the area of residence as a random effect to address clustering [24,25].

Regression assumptions were confirmed via the Shapiro–Wilk test (normality of residuals), the Breusch-Pagan test (heteroscedasticity), and the Durbin-Watson test (independence of observations). Cook’s distances were used to identify influential observations.

Group Differences: To determine differences in the GS across strata defined by qualitative variables (e.g., position played, competition level), Tukey’s test was employed. This test was used to estimate pairwise differences in marginal means, with *p*-values adjusted for multiple comparisons.

General Parameters: All estimates included their corresponding 95% confidence intervals, derived using the delta method. The threshold for statistical significance was set at 5% (α = 0.05). All statistical analyses were performed using R statistical software, version 4.4.0, utilizing packages such as marginal effects [21].

Institutional Review Board Statement: “The study was conducted in accordance with the Declaration of Helsinki and approved by the Provincial Ethics Committee of Granada (CEIm) of the Department of Health and Consumption (Junta de Andalucía, Spain), (protocol code SICEIA-2024-000633, date of approval: 6 June 2024).”

Informed Consent Statement: “Informed consent was obtained from all subjects involved in the study.” For minors (participants under 18s), informed consent was explicitly obtained from their parents or legal guardians, alongside the verbal assent of the participants themselves, who were informed in age-appropriate language about the nature and objectives of the study and their right to withdraw at any time without consequence.

## 3. Results

### 3.1. Anthropometric, Sociodemographic, and Sport-Related Characteristics

#### 3.1.1. Anthropometric Characteristics

The anthropometric characteristics of the participants are presented in Table 1, where the mean age was 13.3 years (SD = 2.2), with a large number of them between 12 and 14 years old. Mean weight and mean height were 55.6 Kg (SD = 15.1) and 163.3 cm (SD = 10.3), respectively. Finally, the participants were in a healthy range for both BMI z-score and % BF.

#### 3.1.2. Socio-Demographic Characteristics and Economic Data

The majority of participants resided in the Beiro area (41.7%). Parental education was notably high, with university education predominating among both fathers (69.4%) and mothers (66.7%). (Table 2).

We observed a significant association between residential area income and ED risk. For every one-unit increase in the logarithm of mean gross income, the Global Score (GS) increased by 2.77 points (beta = 2.77; 95% CI: 0.35–5.17; *p* < 0.05). While the downtown area showed the highest average income (€34,848) and the highest median GS (1.37; IQR: 0.81–3.18), the Beiro area had the lowest average income (€20,354) and a low median GS (0.13; IQR: 0.05–0.49). (Table 3).

The distribution of the Global Score (GS) varied by residence, with areas like Zaidín showing considerable score dispersion, while areas like Beiro and Churriana showed more concentrated and lower scores. The mean scores obtained across the four S-EDE-Q subscales (Restraint, Eating Concern, Shape Concern, and Weight Concern) contribute to the overall Global Score (Figure 1).

#### 3.1.3. Sport-Related Variables

Most players occupied the point guard (30.6%) or pivot (22.2%) positions, with half of the sample (50%) competing at the local level. Players had an average sport experience of 5.2 $ years (SD = 3.3) and dedicated an average of 7.6 h (SD = 3.5) per week to training (Table 4).

### 3.2. Scores Obtained on the Subscales of the S-EDE-Q Questionnaire

Overall, the players scored low on the subscales of the questionnaire, indicating a low overall prevalence of clinically significant eating behaviours and concerns within this sample; however, a small percentage of players showed elevated scores. On the restraint and eating concern subscales, only one participant (2.8%) per subscale exceeded the clinically significant threshold. On the other hand, in the shape concern and weight concern subscales, greater variability was observed, with means of 0.98 (SD = 1.55) and 0.94 (SD = 1.37), respectively, as well as three participants (8.3%) showing clinically significant scores (Figure 2).

### 3.3. Relationship Between Sociodemographic Variables and the Global Score of the S-EDE-Q Questionnaire

The results shown in Figure 3 indicate how the global score of the S-EDE-Q questionnaire varied as a function of different socio-demographic factors. Firstly, a significantly higher mean (3.950) is observed for participants whose parents have primary education, with a confidence interval (CI) between 1.771 and 6.129. In comparison, the other higher levels of education, such as secondary school, university, and graduate school, have lower means. The differences between primary education and the other educational levels are statistically significant: primary–secondary (3.181 [95% CI: 0.006–6.357], *p* < 0.05), primary–university (3.268 [95% CI: 0.311–6.225], *p* = 0.05), and primary–postgraduate (3.543 [95% CI: 0.367–6.718], *p* = 0.05). On the other hand, comparisons between high school, university, and postgraduate showed no significant differences.

Secondly, although no significant differences were found in relation to the mother’s educational level, mothers with secondary education had a higher mean of 1.45 [95% CI: 0.391, 2.509], mothers with university education had a mean of 0.62 [95% CI: 0.137, 1.103], and mothers with postgraduate education had a mean of 0.68 [95% CI: −0.214, 1.575].

Finally, with respect to socio-economic status, significant differences in global scores were observed. The contrast between the ‘lower-middle’ and ‘middle’ conditions showed an estimate of 3.644 (ES = 0.678, 95% CI: 1.806–5.481, *p* < 0.05), while ‘lower-middle’ versus ‘upper-middle’ showed an estimate of 2.959 (ES = 0.693, 95% CI: 1.081–4.838, *p* < 0.05). However, no significant difference was found between the ‘medium-low’ and ‘high’ levels, with an estimate of −1.050. Also, when comparing ‘medium’ and ‘medium-high’, an estimate of −0.685 was observed, while the comparison between ‘medium’ and ‘high’ showed an estimate of −4.694.

### 3.4. Correlation of Anthropometric Variables and the Overall Score of the S-EDE-Q Questionnaire

In the analysis of the average marginal effects (AMGE) of the anthropometric variables, shown in Figure 4, significant relationships are observed between some of them. Age had an AMGE of 0.297 (95% CI 0.101 to 0.492, *p* < 0.05), weight had an effect of 0.040 (95% CI 0.018 to 0.061, *p* < 0.05), and height had the highest effect at 0.0572 (95% CI 2.773 to 8.657, *p* < 0.05). BMI had a significant AMGE of 0.244 (95% CI 0.063 to 0.424, *p* < 0.05), as did % WC, which had an effect of 0.051 (95% CI −0.003 to 0.105, *p* < 0.05), but this did not reach statistical significance. Finally, MM had a significant effect of 0.078 (95% CI 0.048 to 0.109, *p* < 0.05).

### 3.5. Relationship Between Sport-Related Characteristics and the Overall Score in the S-EDE-Q Questionnaire

The corresponding results of the sport-related variables such as playing position, competition level, sport experience, weekly training time, and medical history are presented in detail in Figure 5.

The overall score of the questionnaire varied according to playing position, with players occupying the center position having the highest mean (1.624) with a 95% confidence interval between 0.814 and 2.434, in contrast to players occupying the point guard position, who had a mean of 0.761 (95% CI: 0.070 to 1.452), while point guards, forwards and players without a defined position had a lower overall score of 0.656 (95% CI: −0.489 to 1.802), 0.220 (95% CI: −0.805 to 1.245) and 0.258 (95% CI: −0.608 to 1.124), respectively. Finally, the power forward position showed no significant variability (0.000, 95% CI −2.291 to 2.291).

In relation to the level of competition, significant differences were observed between the different levels, with players competing at the national level having a significantly higher mean overall score (2.663) (95% CI: 1.677, 3.648) compared to those competing at the regional (0.565) (95% CI: 0.038, 1.092) and local (0.463) (95% CI: −0.001, 0.928) levels. Comparisons between local and national (*p* < 0.05) and between regional and national (*p* < 0.05) levels show statistically significant differences, while between local and regional levels, no significant differences were observed (*p* = 0.954).

In sports experience, no significant correlation was found between these two factors (Spearman correlation coefficient = 0.258, *p* = 0.128). Also, the weekly dedication in training hours shows a negative correlation with the overall score, although it is not significant (Spearman correlation coefficient = −0.137, *p* = 0.425).

Finally, female players with a sport-related medical history had a higher mean of 1.073 (95% CI: 0.167, 1.979) compared to those with no medical history, 0.726 (95% CI: 0.256, 1.196), and those with a non-sport-related medical history, 0.167 (95% CI: −1.217, 1.551).

### 3.6. Behavioural Results of the S-EDE-Q Assessment

Based on the S-EDE-Q questionnaire, eating and exercise behavior were further analysed, with the relevant results presented in Table 5.

It was observed that 11.1% of the participants reported some form of eating restriction, although none did so on a regular basis. On the other hand, with regard to any objective binge episodes, 41.7% of the participants reported having experienced at least one episode, with 78.9% of the participants confirming this behaviour in the group that responded. In addition, 16.7% of participants in the total sample reported episodes of any subjective binge, 66.7% of which occurred regularly. Any self-induced vomiting behaviour was less frequent, reported by 5.6% of participants, while any laxative and diuretic misuse was reported by 2.8% of participants in each case. Finally, any excessive exercise was observed in 13.9% of the participants, with 62.5% occurrence in the individual responses.

Regarding habitual behaviour, no regular dietary restraint was observed among participants; however, 13.9% reported regular objective binge episodes, and 16.7% reported regular subjective binge episodes. Regular self-induced vomiting and regular laxative and diuretic misuse remained at 5.6% and 2.8%, respectively. Finally, regular excessive exercise was reported by 5.6% of participants, with 25.0% occurrence in individual responses.

### 3.7. Preliminary Diagnostic Analysis Based on the Results of the S-EDE-Q Questionnaire and the Criteria of the DSM-5 Manual

A preliminary diagnostic analysis of the participants has been carried out based on the results obtained from the S-EDE-Q questionnaire and the diagnostic criteria of the DSM-5 manual. It is crucial to stress that, as a screening tool, the risk classification obtained via the S-EDE-Q does not constitute a definitive clinical diagnosis and does not replace the comprehensive evaluation required through a semi-structured diagnostic interview (such as the EDE) in accordance with DSM-5 criteria.

The findings for each participant with elevated scores are presented below:Participant: AT19GR

BMI: 21.6 (Normal).

High score (S-EDE-Q Questionnaire): Restraint.

Habitual behaviours:-Any dietary restraint.-Any objective binge episode.-Any subjective binge episode.-Regular objective binge episodes, regular objective binge episodes.-Regular subjective binge episodes, regular subjective binge episodes).

Diagnostic criteria according to the DSM-5 manual: Given that she presents regular episodes of binge eating without compensatory behaviours, it could be considered a diagnosis of AT.

2.Participant: AT25GR

BMI: 25.0 (overweight).

High score (S-EDE-Q questionnaire): Shape concern, weight concern and Global score.

Habitual behaviours:-Any dietary restraint, any dietary restraint.-Any objective binge episode.-Any laxative misuse episode, any laxative misuse.-Regular objective binge episodes.-Regular laxative misuse episodes.

Diagnostic criteria according to DSM-5: The combination of regular binge eating and laxative use suggests a diagnosis of BN.

3.Participant: AT33GR

BMI: 30.7 (Obesity type I)

High score (S-EDE-Q questionnaire): Eating concern, shape concern, weight concern and global score.

Habitual behaviours:-Any dietary restraint, any dietary restraint.-Any objective binge episode.-Any subjective binge episode, any subjective binge episode.-Any self-induced vomiting episode, any self-induced vomiting.-Any episode of laxative misuse, any laxative misuse.-Any episode of diuretic misuse, any diuretic misuse.-Regular objective binge episodes.-Regular subjective binge episodes, regular subjective binge episodes.-Regular self-induced vomiting episodes, regular self-induced vomiting.-Regular diuretic misuse, regular diuretic misuse.-Regular episodes of excessive exercise, regular excessive exercise.

Diagnostic criteria according to DSM-5: The presence of these compensatory behaviours together with preoccupations and episodes of binge eating indicates a diagnosis of BN.

## 4. Discussion

The aim of this exploratory study was to evaluate the presence of eating disorders in women aged 10 to 18 years from a high-level basketball club in Granada, Spain. The research focused on assessing not only the risk of ED but also the presence of confirmed cases. In addition, body composition, eating behaviors, sociodemographic, and sporting characteristics were analyzed.

Our study found that the mean S-EDE-Q scores were generally low, yet 8.4% of the participants were identified as being at risk for an Eating Disorder (ED). Only 2.8% surpassed the threshold for dietary restraint and preoccupation with food, while 5.6% exceeded the thresholds for preoccupation with shape and weight.

These findings, which point to a lower but present risk in female basketball players, align closely with prior research on this specific team sport. For instance, Michou and Costarelli (2011) similarly found that 11% of female basketball players displayed disordered eating attitudes [6]. This contrasts with the much higher rates, reaching up to 45%, often reported in aesthetic and endurance sports [14,26,27]. Our results corroborate the notion that, while team sports like basketball exhibit lower rates, a significant risk remains [14].

Risk Predictors and Vulnerability Periods: We found a significant correlation between older age and higher ED symptoms. With a mean age of 13.3 years (SD = 2.2) in our sample, this emphasizes that the mid-adolescent period is a critical window for the emergence of ED symptoms, consistent with literature indicating the peak prevalence of Anorexia Nervosa (AN) between 13 and 18 years [28] and recognizing adolescence as a heightened period of vulnerability [14,29].

Furthermore, several specific factors were identified as predictors: A higher BMI z-score correlated with a higher ED risk, a lower educational level correlated with a higher ED risk, and National-level players and those in the pivot position showed a significantly higher risk compared to local or regional players. A history of sports-related medical problems was also associated with a higher risk of ED.

The finding that the majority of our participants (77.8%) had a BMI z-score within normal limits, despite the identified risk factors, corroborates studies like Kontele et al. (2022) [7]. This underscores the critical need to utilize symptom-based screening tools like the S-EDE-Q, as risk is not reliably indicated by body weight alone in this athletic population.

Regarding parental education, participants whose parents had only a primary school education exhibited significantly higher mean global scores on the questionnaire compared to those with higher educational levels (university or postgraduate). While ED can affect individuals across all socioeconomic backgrounds, lower socioeconomic status is often linked to a higher incidence of compulsory treatment for anorexia nervosa [30]. Conversely, higher educational attainment, combined with factors like body image concerns and the use of appearance-oriented social media, is associated with an increased risk of exhibiting ED symptoms [31]. Additionally, parental education is generally linked to healthier eating habits, as adolescents with well-educated parents tend to consume more fruits and vegetables [32]. Furthermore, family dynamics, including communication and cohesion, act as either protective or risk factors for maladaptive eating behaviors in adolescents [10,26].

Anthropometric Predictors: Weight, height, and BMI all exhibited marginally significant effects on the Global Score. Our findings concerning weight are consistent with previous research showing that higher body weight can contribute to increased body dissatisfaction, thereby elevating ED risk [8,33]. Height dissatisfaction has also been correlated with body dissatisfaction and the pursuit of thinness or muscularity, further influencing ED development [34].

The BMI z-score results align with existing literature, which identifies a higher BMI as a significant predictor of ED, demonstrating an association with elevated global assessment scores [8,34]. However, these findings require careful interpretation. Borowiec et al. (2023) emphasize that the impact of BMI should be considered alongside factors like body satisfaction and the type of sport [35]. Furthermore, many studies supporting this BMI-ED association have focused predominantly on male populations, highlighting a potential gender bias that future research must address [15].

Regarding body fat percentage (% BF), a non-significant trend suggested a potential association with the risk of developing atypical eating behaviors. Factors such as rigorous weight assessments, body composition pressures, coach expectations, and public exposure of results may trigger such behaviors, though influences vary by sport [2,15,36]. Finally, muscle mass (MM) demonstrated a significant effect, underscoring its importance in shaping body image perceptions and ED risk [2,16].

Sport-Specific Predictors: Our results identified that National-level players and those in the pivot position had the highest mean scores, indicating a heightened risk. While research specifically linking basketball playing positions and ED is limited, the literature on male team sports suggests that team participation can heighten concerns about weight gain and food intake, increasing ED likelihood [37,38]. Other identified sport-specific risk factors, such as early specialization, injuries, frequent weight management, and dietary practices, may also render athletes more susceptible to these disorders [14].

Our data revealed that national-level players reported significantly higher mean S-EDE-Q scores compared to their regional and local counterparts. This aligns with findings by Teixidor-Batlle et al. (2021) [39], who identified the level of competition as a potential ED risk factor. The intensification of performance expectations, weight-related pressures from coaches and teammates, and uniform requirements often accompany higher competitive levels across various sports [39]. However, it is also important to consider the nuanced perspective that higher-level athletes may benefit from enhanced psychological support, which could potentially mitigate risk [39].

The observation of increased risk at the national level supports the idea that, even in team sports, the drive for elite performance can supersede the protective aspects of the sport itself. This echoes Kampouri et al.’s (2019) finding that elite female team athletes may exhibit eating behaviors similar to those in individual sports, highlighting that performance pressure is a key driver, irrespective of the sport type [34].

We found that players in the pivot position had the highest mean overall scores, indicating a heightened risk. While specific research linking basketball positions to ED risk is scarce, this finding suggests that the unique physical demands or body type expectations associated with the pivot role may contribute to greater psychological vulnerability regarding body image and weight.

Additionally, female athletes with a history of sport-related medical issues (e.g., injuries) reported significantly higher mean overall scores, positioning injury as a key risk factor. Previous research supports this, showing that athletes with injury histories often experience increased concerns about their weight and body shape, driven by the need to maintain athletic performance despite physical limitations [14].

In contrast to the risks identified above, our study found no significant correlation between sports experience or weekly training commitment and overall scores. This suggests that dedication to the sport itself might serve as a protective factor, possibly by enhancing mood and self-esteem [36]. Nonetheless, it is acknowledged that other sport-specific factors—such as the age training begins, emphasis on weight control, and environmental pressures—are known to increase ED risk in the athletic population [2,14,35].

Finally, while 11.1% of participants reported some form of dietary restriction (DR), this behavior was not consistently maintained by any player. This aligns with existing literature suggesting that DR is a prevalent practice among athletes aiming to enhance performance or maintain a specific body image [14,16], underscoring the general athlete tendency to self-evaluate based on body shape and food control [37].

Furthermore, 41.7% of participants reported experiencing at least one objective overeating episode (Any OBE), with 78.9% of these individuals confirming that such behavior occurred regularly. In contrast, subjective overeating episodes (Any SBE) were reported by 16.7% of participants, with 66.7% indicating that these episodes occurred frequently. Both objective and subjective overeating episodes serve as warning signs, as they are associated with an elevated risk of developing ED and can negatively impact sports performance. This is consistent with studies that document a high prevalence of these behaviors among athletes facing intense competitive pressure, where concerns about food, body shape, and weight often lead to increased dietary restrictions, thereby heightening the risk of overeating or binge eating [14,24,32,37]. Marí-Sanchis et al. (2022) [2] note that dietary restrictions, when combined with purging behaviors and fluid intake alterations, can impair athletes’ concentration, coordination, and emotional well-being, while also increasing the risk of injury.

Self-induced vomiting behaviors (any SIV) were reported by 5.6% of participants, while the use of laxatives and diuretics (any LAX and any IUD) was noted in 2.8% of participants. These findings are consistent with research on aesthetic sports, where such behaviors are linked to heightened dietary restrictions and excessive exercise, particularly among female athletes, who exhibit a higher prevalence of ED compared to their male counterparts [34,38]. Regarding excessive exercise (any EX), 13.9% of participants reported engaging in this behavior, with 62.5% indicating that it occurred regularly. Excessive exercise is recognized as a potential trigger for ED, as overtraining can lead to significant caloric deficits, creating a psychological or biological environment conducive to the development of ED [17]. While often perceived as a demonstration of commitment to the sport, excessive exercise can result in injuries, chronic fatigue, and other health complications if not managed appropriately [15,17].

These findings suggest that, although the prevalence of extreme eating behaviors is not alarmingly high, a notable subset of female players engages in risky behaviors. These diagnoses highlight the diversity and severity of ED that can manifest in athletes, underscoring the necessity for targeted prevention and treatment strategies.

Practical Implications: The athletic environment plays a critical role; coaches are often seen as authorities on sports, nutrition, and weight management [14,40,41]. Their comments, along with pressures from teammates and parents, can significantly exacerbate body image and weight concerns [14,35].

Coaches are in regular contact with athletes and are frequently the first to observe subtle changes in weight, eating behavior, body composition, or mood. However, a lack of knowledge may lead them to unintentionally promote risky behaviors, which can trigger or exacerbate EDs and delay crucial detection [42].

It is vital that coaches receive specific and mandatory training on the physical (e.g., excessive weight loss, fatigue) and behavioral (e.g., restraint, binge eating, excessive preoccupation) signs and symptoms of EDs [43].

Coaches must be equipped to foster dialogue with parents and health professionals and establish clear protocols for when and how to refer an athlete to a qualified nutritionist, psychologist, or physician specializing in EDs [44].

While height and weight measurements are necessary for monitoring performance, coaches must execute them with extreme discretion and professionalism in an environment that minimizes anxiety, especially for athletes predisposed to ED [40].

Although EDs are more often associated with sports emphasizing leanness and aesthetics [2,6,12,15], vigilant monitoring is essential across all elite sports, including basketball, for early diagnosis.

Problems tend to escalate when dietary practices are unsupervised. Inadequate guidance on weight management, rather than merely the directive to lose weight, has been shown to precipitate EDs [12,17]. Therefore, all weight and nutrition directives must be supervised by a qualified sports nutritionist.

Coaches should prioritize health and performance markers over arbitrary weight targets, shifting the environment away from a sole focus on body size as a measure of athletic worth.

The clinical relevance of our study is of utmost importance for the prevention of ED within the sports club setting. We provide robust justification for the mandatory implementation of preventative strategies, such as positive, performance-focused nutrition education programs, training technical staff for early warning sign detection, and establishing clear club policies against weight-based criticism and body shaming. Our data offers clubs a clear roadmap to create a safer and healthier environment for their athletes, protecting them from the psychosocial risks associated with ED.

This study is not without limitations, some of which align with those identified in similar research. First, our sample size is relatively small and may have resulted in low statistical power, which may limit the generalizability of our results to the broader population of female basketball players in Granada and other contexts, including different socioeconomic backgrounds and types of sports. However, our study included all women from the G + B club, so it was not possible to recruit more participants at the time of the study. Previous studies conducted with members of the same sports club have included 15 triathletes [45], 42 cyclists [46], 27 Reserve Officers’ Training Corps cadets [47], 26 dancers [48], or 36 elite competitive adolescent figure skaters [49].

Secondly, as noted by Kontele et al. [7], Ravi et al. [16], and Tafà et al. [26], the cross-sectional design of our study restricts our ability to draw causal inferences, thereby limiting our capacity to assess long-term changes and confining our findings to observational associations. Moreover, while our exploratory approach allowed us to identify several potentially relevant associations, we acknowledge the increased risk of type I error inherent in multiple comparisons. Future research will employ a more focused question and pre-specified hypotheses, derived from these pilot findings, and will implement adequate correction procedures to minimize false positives.

Additionally, similar to the studies conducted by Ravi et al. [16], Tafà et al. [26], and Thompson et al. [50], our research relied on self-reported data, specifically the S-EDE-Q questionnaire. This reliance may compromise the reliability of our findings due to potential inaccuracies in responses, a common limitation in studies utilizing self-reporting tools. Such tools are susceptible to biases, including self-selection bias [16], and may not accurately reflect the assessments that a trained professional would derive through direct interviews [14,43,44]. Another significant limitation, shared with the research of Michou and Costarelli [6] and Ravi et al. [16], is the absence of a control group. This omission hinders our ability to compare the results from our sample with those of a non-sporting population, emphasizing the necessity for greater diversity in sample populations and the inclusion of appropriate control groups to enhance the generalizability and external validity of the findings [6,43].

Another important limitation of this study is the use of convenience sampling, which may introduce selection bias and limit the generalizability of the findings beyond the specific population studied. Given the exploratory and pilot nature of the research, this non-probabilistic sampling approach was selected based on logistical feasibility and access to the target group; however, future studies should aim to incorporate probabilistic sampling strategies to enhance representativeness and external validity.

Finally, as highlighted by Michou and Costarelli [6], there is a notable lack of targeted research on female athletes, which limits our comprehensive understanding of the impact of ED within this demographic. This underscores the importance of considering gender differences and sport-specific factors in research on ED, as well as the scarcity of studies addressing eating attitudes in team sports.

Future research must expand the scope beyond this pilot study to confirm our findings and enhance generalizability. Studies should encompass a broader range of sports populations across diverse geographical areas. This will facilitate a more nuanced comparison of how varied socioeconomic and residential contexts influence ED risk.

To capture the dynamic nature of ED risk, future work should involve multiple S-EDE-Q assessments across an entire sports season. This approach is crucial for understanding how training load and competition dynamics affect eating behaviors and body image perceptions over time. To achieve a comprehensive understanding of contributing factors, research methodology should be enriched by incorporating individual qualitative interviews (as recommended by Bratland-Sanda [14] and Teixidor-Batlle et al. [39]).

The identification of specific predictors (e.g., older age, pivot position, lower parental education) implies a need for targeted prevention strategies. These findings underscore the importance of early, context-specific screening and intervention programs designed to protect the physical and psychological health of vulnerable adolescent female athletes.

Practical Applications: Our study can promote primary prevention of ED by promoting education, fostering a positive body image, and raising awareness among stakeholders to recognize risk factors. Studies of this nature can be useful for understanding the incidence and risk factors in a vulnerable population (young female athletes), as well as for protecting their physical and mental health, preventing the onset of these disorders, facilitating effective recovery, and promoting sustainable and healthy athletic performance [51]. In addition, we provide specific screening tools to identify athletes at risk of developing ED.

## 5. Conclusions

The present pilot study suggests a low overall prevalence of ED symptoms among this cohort of female adolescent basketball players. Crucially, however, the presence of a small subgroup (8.4% of the sample) with clinically significant indicators underscores that a genuine risk exists even within team sports not traditionally associated with pressures for thinness. The observed associations—such as higher ED scores among older players, those in the pivot position, and participants with lower parental educational levels—are strictly preliminary. Due to the exploratory nature of this investigation and its limited sample size, these findings must be interpreted with caution. They are intended primarily to inform the design of future studies.

To confirm these initial findings and rigorously explore causal relationships, future research is necessary that utilizes. Larger, more representative samples with pre-specified hypotheses. Longitudinal designs are particularly valuable for assessing changes in ED risk over the course of an athlete’s development and competitive career.

These research efforts are essential to build an evidence base that can effectively inform and guide monitoring, prevention, and intervention strategies, ultimately protecting the physical and psychological health of female adolescent athletes.

## Figures and Tables

**Figure 1 nutrients-17-03547-f001:**
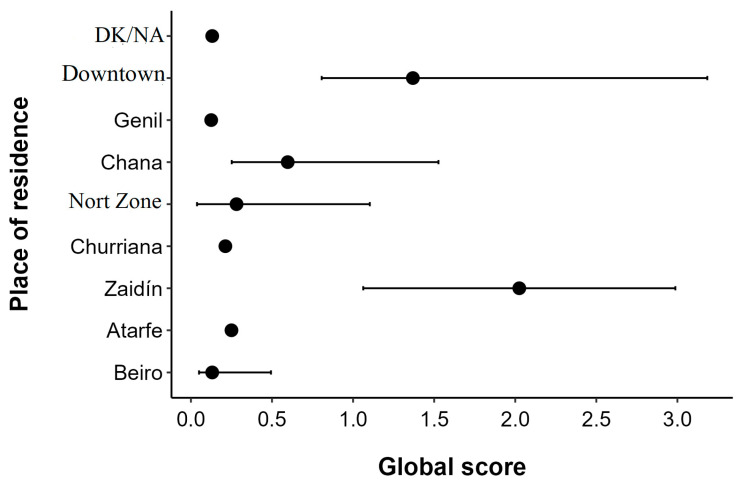
Distribution of the Eating Disorder global score according to participants’ place of residence. Point: This is the mean value or average score observed for the “Global score” variable within that specific group of “Place of residence.” Line: Confidence interval.

**Figure 2 nutrients-17-03547-f002:**
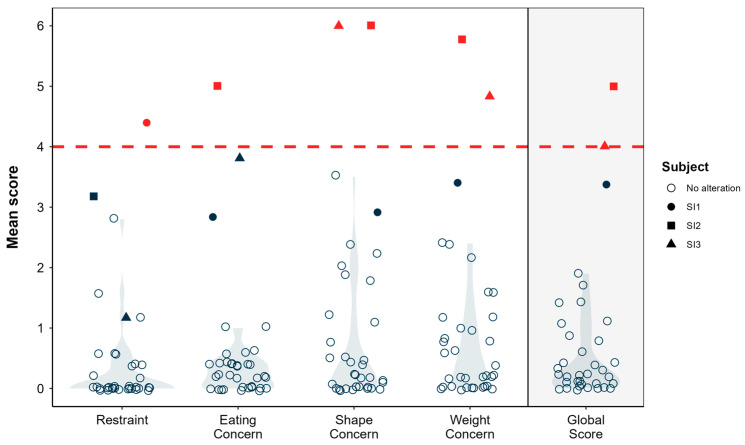
Eating Disorder Subscale Scores. The red shapes (squares and triangles) represent the mean scores of individual subjects identified as S11, S12, and S13. Red Squares (S11), Red Circles (S12), Red Triangles (S13). The horizontal red dashed line at a “Mean score” of 4 is a threshold or cut-off point typically used to indicate a level of concern. The light grey, shaded areas behind the white circles are violin plots. These plots illustrate the distribution of “Mean scores” for the “No alteration” group (represented by the white circles) across each subscale and the Global Score.

**Figure 3 nutrients-17-03547-f003:**
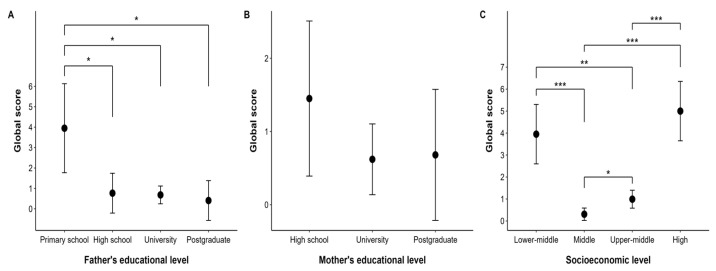
Global Score (GS) of the S-EDE-Q According to Socio-demographic. This figure (**A**–**C**) displays the relationship between the “Global score” and various socioeconomic and educational factors. Each subfigure uses mean plots with 95% confidence intervals to show the average global score for different categories. Note: Asterisks indicate levels of statistical significance: * *p* < 0.05; ** *p* < 0.01; *** *p* < 0.001.

**Figure 4 nutrients-17-03547-f004:**
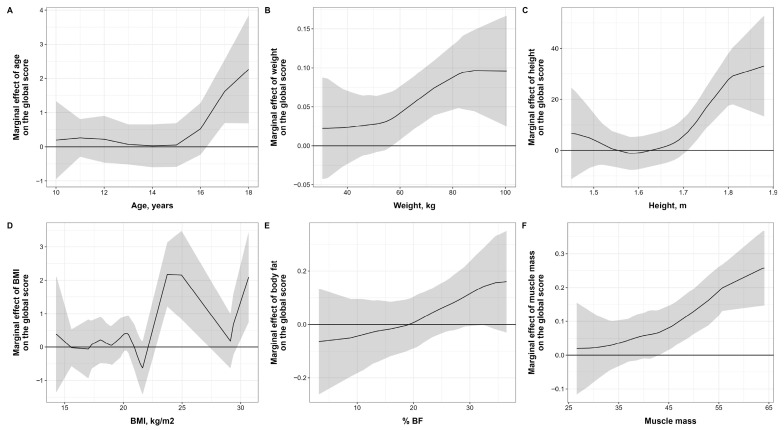
Average Marginal Effect (AMGE) of Anthropometric Variables on the S-EDE-Q Global Score (GS). This figure (**A**–**F**) shows marginal effects graphs. Note: Shaded bands represent the 95% confidence interval. The solid black line in each plot represents the estimated marginal effect of the predictor variable on the “Global score.” The grey shaded area surrounding the solid black line represents the 95% confidence interval for the marginal effect.

**Figure 5 nutrients-17-03547-f005:**
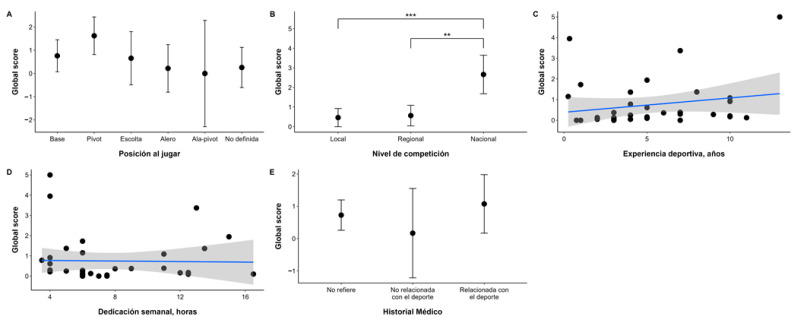
Relationship between sport variables and global score of the S-EDE-Q questionnaire. This figure to explore the relationship between the “Global score” and various factors related to sports participation and medical history. (**A**): Global score by Playing Position. (**B**): Global score by Competition Level. (**C**): Global score by Sports Experience (years). (**D**): Global score by Weekly Dedication (hours). (**E**): Global score by Medical History. The vertical lines extending from the top of the bars (or from the center of the black dots) are the error bars. Subfigures (**C**,**D**): The gray area around a line represents the 95% Confidence Interval for the regression line/trend itself. ** *p* < 0.01; *** *p* < 0.001.

**Table 1 nutrients-17-03547-t001:** Anthropometric characteristics of the participants.

Variables	Ranges	*n* = 36 ^1^ (%)
Age (years)	<12	8.0 (22.2)
	12–14	11.0 (30.6)
	14–16	10.0 (27.8)
	≥16	7.0 (19.4)
Mean (SD)	13.3 (2.2)	
Weight (Kg)	<45	6.0 (16.7)
	45–55	15.0 (41.7)
	55–65	7.0 (19.4)
	≥65	8.0 (22.2)
Mean (SD)	55.6 (15.1)	
Height (cm)	<160	12.0 (33.3)
	160–165	6.0 (16.7)
	165–170	10.0 (27.8)
	≥170	8.0 (22.2)
Mean (SD)	163.3 (10.3)	
BMI z-score	Underweight: <−2	0.0 (0.0)
	Normal weight: ≥−2 y < +1	26.0 (77.8)
	Overweight: ≥+1 y < +2	6 (16.7)
	Obesity: ≥+2	4.0 (11.1)
Mean (SD)	0.42 (1.04)	
Body Fat (%)	<20	10.0 (27.8)
	20–30	20.0 (55.6)
	≥30	6.0 (16.7)
Mean (SD)	23.3 (7.5)	
Lean mass (Kg)	<35	7.0 (19.4)
	35–45	20.0 (55.6)
	≥45	9.0 (25.0)
Mean (SD)	41.9 (8.7)	

^1^ *n* (%). BMI z-score: Body Mass Index z-score according to the WHO (n.d.) SD: Standard Deviation. No significant differences in any of the variables.

**Table 2 nutrients-17-03547-t002:** Socio-demographic characteristics of participants.

Variables	Stratum	*n* = 36 (%)
Father’s educational level	Primary School	1.0 (2.8)
	Secondary	5.0 (13.9)
	University	25.0 (69.4)
	Postgraduate	5.0 (13.9)
Mother’s educational level	Secondary	5.0 (13.9)
	University	24.0 (66.7)
	Postgraduate	7.0 (19.4)
Socio-economic level Mean (SD)	Lower-middle	1.0 (2.8)
	Medium	23.0 (63.9)
	Upper-middle	11.0 (30.6)
	High	1.0 (2.8)

**Table 3 nutrients-17-03547-t003:** Economic data by residence.

Residential Area	Average Gross Income [22]	Number of Participants *n* (%)	Median Global Score (IQR)
Beiro	€20,354	15.0 (41.7)	0.13 (0.05–0.49)
Atarfe	€20,746	1.0 (2.8)	0.25 (0.25–0.25)
Zaidín	€22,038	2.0 (5.6)	2.02 (1.06–2.99)
Churriana	€22,109	1.0 (2.8)	0.21 (0.21–0.21)
North Zone	€22,559	8.0 (22.2)	0.28 (0.04–1.10)
Chana	€23,291	4.0 (11.1)	0.60 (0.25–1.53)
Genil	€30,181	1.0 (2.8)	0.12 (0.12–0.12)
Downtown	€34,848	3.0 (8.3)	1.37 (0.81–3.18)
DK/NA	-	1.0 (2.8)	0.13 (0.13–0.13)

**Table 4 nutrients-17-03547-t004:** Sport-related variables.

Variables		*n* = 36 (%)
Playing position	Point Guard	11.0 (30.6)
	Pivot	8.0 (22.2)
	Shooting guard	4.0 (11.1)
	Small forward	5.0 (13.9)
	Power forward	1.0 (2.8)
	Not defined	7.0 (19.4)
Competition level	Local	18.0 (50.0)
	Regional	14.0 (38.9)
	National	4.0 (11.1)
Sport experience (years)	<3	7.0 (19.4)
	3–7	17.0 (47.2)
	≥7	12.0 (33.3)
Mean (SD)		5.2 (3.3)
Training per week (hours)	<5	7.0 (19.4)
	5–10	20.0 (55.6)
	≥10	9.0 (25.0)
Mean (SD)		7.6 (3.5)
Medical History	Not related to Sport	3.0 (8.3)
	Sport related	7.0 (19.4)
	Does not refer	26.0 (72.2)

**Table 5 nutrients-17-03547-t005:** Frequency of Eating and Exercise Behaviour.

Behaviour	Occurrence/Responses	Parcial Proportion ^1^	Total Proportion ^2^
Any DR: Any dietary restraint	4/36	11.1%	11.1%
Any OBE: Any objective binge episode	15/19	78.9%	41.7%
Any SBE: Any subjective binge episode	6/9	66.7%	16.7%
Any SIV: Any self-induced vomiting	2/2	100.0%	5.6%
Any LAX: Any laxative misuse	1/1	100.0%	2.8%
Any DIUR: Any diuretic misuse	1/1	100.0%	2.8%
Any EX: Any excessive exercise	5/8	62.5%	13.9%
Reg DR: Regular dietary restraint	0/36	0.0%	0.0%
Reg OBE: Regular objective binge episodes	5/19	26.3%	13.9%
Reg SBE: Regular subjective binge episodes	6/9	66.7%	16.7%
Reg SIV: Regular self-induced vomiting	2/2	100.0%	5.6%
Reg LAX: Regular laxative misuse	1/1	100.0%	2.8%
Reg DIUR: Regular diuretic misuse	1/1	100.0%	2.8%
Reg EX: Regular excessive exercise	2/8	25.0%	5.6%

^1^ proportion calculated from the total number of people who responded to that item. ^2^ proportion of the total number of participants.

## Data Availability

The data presented in this study are available on request from the corresponding author due to confidentiality and privacy restrictions in accordance with ethical guidelines and data protection laws.

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
