# Peer review of "Prevalence and Risk Factors for Eating Disorders in Adolescent Female Basketball Players"

_nutrients, 2025, doi:10.3390/nu17223547_

Round 1

Reviewer 1 Report

Comments and Suggestions for Authors

REPORT REVIEW (nutrients-3942626)
Prevalence and Risk Factors of Eating Disorders in Female Adolescent Basketball Players: A Pilot Study.

ABSTRACT
Please reduce excessive numerical results, keeping only the most relevant ones, and include "This cross-sectional pilot study..." in the methods.

INTRODUCTION

- Revise the introduction to summarize the European epidemiological data without synthesis. Furthermore, make the research hypotheses more explicit.
- What theoretical mechanisms justify the expectation of differences between the center and point guard positions? Weight pressure? Body composition?

METHODS
- In my opinion, the sample size calculation via GPower* is not consistent with a pilot study; it should be described only as "estimation of minimal detectable effect." I suggest deleting it.
- Please indicate how the questionnaire was administered, whether self-administered or supervised. - Please note: The authors need to moderate the text, as the term "DSM-5 criteria" as a dependent variable is imprecise, as the study used the S-EDE-Q, not a clinical diagnosis.
- Was the questionnaire administered before or after training? In what setting?
- Was there any procedure to control response bias, such as complete anonymity, enhanced confidentiality?
RESULTS
- The SMA for height (5.72) seems disproportionate. Is the value correct?
- The preliminary DSM-5 diagnosis (lines 450–496) is risky: the S-EDE-Q does not replace a diagnostic interview.

DISCUSSION
- It is very long. The authors repeat almost all the numerical results; the narrative could focus on sociodemographic, anthropometric, and athletic factors.
- It seems to me that the discussion of the role of coaches would be better under a subtitle "Practical Implications." - How do the authors interpret the paradoxical finding that height and muscle mass are associated with a higher risk score? Is it a statistical effect (collinearity with age/weight)? This needs to be better explained.
- Avoid repeating data already reported in the results.
- The authors should highlight the clinical relevance for prevention in clubs.

CONCLUSION
- Revise to reduce it to a paragraph that summarizes the identified risk and the need for larger studies.

Author Response

Comments 1: [Please reduce excessive numerical results, keeping only the most relevant ones, and include "This cross-sectional pilot study..." in the methods].

Respone 1: Thank you very much for your comment. The paragraph has been summarized and less important numbers have been removed, and in methods we added the text “This cross-sectional pilot study examined 36 participants”.

INTRODUCTION

Comments 2 – [Revise the introduction to summarize the European epidemiological data without synthesis. Furthermore, make the research hypotheses more explicit].

Response 2: Thank you very much for your comment. The paragraph has been rewritten to include European statistics. (Lines 58-74) and included more specific hypotheses (Lines 129- 137).

Comments 3- [What theoretical mechanisms justify the expectation of differences between the center and point guard positions? Weight pressure? Body composition?]

Response 3: That is an excellent question that probes the theoretical and causal mechanisms behind the observed Eating Disorder (ED) risk differences between the Center and Point Guard positions.

The justification lies primarily in the distinct biomechanical demands of each role, which lead to different body composition targets and, consequently, different types of body image pressure.

The Center position inherently demands greater body size, higher absolute weight, and significantly greater muscle mass (MM) for effective rebounding, physical contact, and post-play.

The Point Guard position prioritizes speed, agility, quickness, and endurance. While they are not expected to be extremely thin, the demands favor a leaner physique.

Precisely, one of our conclusions is that the Center position carries a high risk and suggests that the pressure to achieve and maintain a large, muscular body (pressure for body composition) and the resulting conflict with society's aesthetic ideals are powerful factors that increase the risk of ED, which could outweigh the more general pressure to be thin felt by point guard (lines 687- 689).

METHODS

Comments 4- [In my opinion, the sample size calculation via GPower* is not consistent with a pilot study; it should be described only as "estimation of minimal detectable effect." I suggest deleting it].

Response 4: Thank you very much for your suggestion. Lines 147-148 have been rewritten.

Comments 5- [Please indicate how the questionnaire was administered, whether self-administered or supervised. - Please note: The authors need to moderate the text, as the term "DSM-5 criteria" as a dependent variable is imprecise, as the study used the S-EDE-Q, not a clinical diagnosis].

Response 5: Thank you very much for your comment. Lines 168-169 indicate that the questionnaire was administered under supervision.

Line 212 rewrites the dependent variables (obtained using S-EDE-Q).

Comments 6- [Was the questionnaire administered before or after training? In what setting?]

Response 6: The questionnaire was administered before the scheduled training session.

To ensure the validity of self-reported data, the environment needed to be quiet to minimize distractions and private to ensure anonymity and honesty, especially when addressing sensitive topics such as eating habits. Therefore, the questionnaire was administered in a private, quiet, and supervised environment within the club's facilities (a meeting room).

The players were given sufficient time on a normal training day in the team's technical meeting room. The researcher KVF provided explanations and BCM assisted her in the session.

Comments 7- [Was there any procedure to control response bias, such as complete anonymity, enhanced confidentiality?]

Response 7: Thank you very much for your comment. The main procedure mentioned in our text for controlling response bias and ensuring confidentiality was anonymization: Lines 182- 183.  

RESULTS

Comments 8- [The SMA for height (5.72) seems disproportionate. Is the value correct?]

Response 8: Thank you very much for your observation. Line 342 has been corrected. In the linear regression model (AME) used, we mistakenly entered meters instead of centimeters. Using this measurement, an AME of 0.0572 per centimeter is much more reasonable.

Comments 9- [The preliminary DSM-5 diagnosis (lines 450–496) is risky: the S-EDE-Q does not replace a diagnostic interview].

Response 9: Thank you very much for your observation. We have introduced lines 404-407 to clarify that concept: “It is crucial to stress that, as a screening tool, the risk classification obtained via the S-EDE-Q does not constitute a definitive clinical diagnosis and does not replace the comprehensive evaluation required through a semi-structured diagnostic interview (such as the EDE) in accordance with DSM-5 criteria”

DISCUSSION

Comments 10- [It is very long. The authors repeat almost all the numerical results; the narrative could focus on sociodemographic, anthropometric, and athletic factors].

Response 10: Thank you very much for your observation. We have rewritten the discussion (lines 456- 554), removing an excessive amount of numerical data and focusing on sociodemographic, anthropometric, and sporting factors.

Comments 11- [It seems to me that the discussion of the role of coaches would be better under a subtitle "Practical Implications." - How do the authors interpret the paradoxical finding that height and muscle mass are associated with a higher risk score? Is it a statistical effect (collinearity with age/weight)? This needs to be better explained].

Response 11: Thank you very much for your observation. We have included a section entitled "Practical Implications" (Lines 584-611).

On the other hand, we believe we attempted to answer this question in comment 3. From our perspective, the center position inherently demands greater body size, higher absolute weight, and significantly greater muscle mass (MM) for effective rebounding, physical contact, and post-play recovery.

Comments 12- [Avoid repeating data already reported in the results].

Response 12: Thank you very much for your observation. We have removed the most numerical data from the discussion.

Comments 13- [The authors should highlight the clinical relevance for prevention in clubs].

Response 13: Thank you very much for your observation. In the lines 612-618 we intend to include this observation.

CONCLUSION

Comments 14- [Revise to reduce it to a paragraph that summarizes the identified risk and the need for larger studies].

Response 14: Thank you very much for your observation. We have summarized the conclusions (Lines 683-691).

We have also written lines 692-695 indicating the need for studies with a larger number of participants.

Reviewer 2 Report

Comments and Suggestions for Authors

General Comments
This manuscript addresses an important and sensitive topic: the prevalence and risk factors of eating disorders among adolescent female basketball players. The study has a clear structure, with well-defined objectives, a rigorous methodological approach, and appropriate statistical analyses for a pilot design. The introduction is informative, and the results are presented in sufficient detail. However, several sections require clarification, improved conciseness, and better alignment between methods, results, and discussion. There are also multiple points of redundancy, formatting inconsistencies, and places where the language could be more precise and scientific.

Specific comments
Title and Abstract
- Lines 2–4: Simplify the title to improve readability (e.g., remove or rephrase 'Pilot Study' at the end or specify more clearly the sample and context).
- Lines 13–34: Consider including effect sizes and confidence intervals for all major results in the abstract.
- Lines 22–29: Rephrase long and complex sentences to make the key findings more direct and impactful.

Introduction
- Lines 37–60: Merge repetitive sentences describing ED definitions and historical context. Avoid excessive detail not directly relevant to the study aim.
- Lines 61–77: Clarify transitions between global prevalence data and basketball-specific context to improve logical flow.
- Lines 118–134: Condense background justification into more concise statements directly linked to the study objectives.

Materials and Methods
- Lines 138–148: Rephrase sentences to reduce redundancy around sample recruitment.
- Lines 163–212: Streamline the description of the S-EDE-Q instrument; focus on its relevance, subscales, and cut-off scores rather than repeating definitions.
- Lines 213–239: Anthropometric procedures could be shortened by removing step-by-step measurement details not essential for replication.
- Lines 260–296: Specify statistical methods more succinctly; clearly indicate which tests were used for each variable.
- Ensure consistent reporting format for p-values (e.g., 'p < 0.05', 'p = 0.031').

Results
- Lines 309–341: Simplify the description of anthropometric and socio-demographic characteristics. Remove narrative repetition of table content.
- Table 2 and Table 3: Standardize formatting of percentages and decimal points.
- Lines 357–397: Ensure clarity in figure captions and statistical terms (e.g., 'AME' should be defined upon first use).
- Figures 3–5: Consider simplifying figure legends to highlight main findings only.

Discussion
- Lines 497–534: Improve alignment between the main findings and the literature cited; avoid overgeneralization.
- Lines 539–577: Some sentences are excessively long and could be split for clarity.
- Lines 579–614: Clarify the interpretation of competition level and position-related findings; ensure statements are supported by the data.
- Lines 648–666: Strengthen recommendations regarding early detection strategies and the role of coaches.
- Improve coherence between the descriptive and analytical findings and the conclusions drawn.

Conclusion
- Lines 713–720: Rephrase for conciseness and impact; clearly highlight key implications for practice and future research.
- Emphasize the exploratory nature of the study and the need for larger longitudinal research.

References and Formatting
- Ensure uniform formatting of references throughout (capitalization, punctuation, DOI formatting).
- Verify consistency in citation numbering and in-text referencing.
- Remove redundant or repeated references if present.

Author Response

Comments 1: [Lines 2–4: Simplify the title to improve readability (e.g., remove or rephrase 'Pilot Study' at the end or specify more clearly the sample and context).]

Response 1: Thank you for pointin this out. We agree with this comments. We have summarized the title.

Comments 2: [Lines 13–34: Consider including effect sizes and confidence intervals for all major results in the abstract.].

Response 2: Only the most important results indicating effect sizes and confidence intervals have been written. Lines 21-23 have been removed.

Comments 3: [Lines 22–29: Rephrase long and complex sentences to make the key findings more direct and impactful.

Response 3: We have rewritten the conclusion to highlight the most direct and impactful key findings (Lines 28-31).

Comments 4: [Lines 37–60: Merge repetitive sentences describing ED definitions and historical context. Avoid excessive detail not directly relevant to the study aim].

Response 4: Thank you very much for your feedback. We have rewritten that paragraph. Repetitive phrases have been merged, and excessive detail has been removed. Repetitive phrases have been merged, avoiding excessive detail (lines 31-57).

Comments 5: [Lines 61–77: Clarify transitions between global prevalence data and basketball-specific context to improve logical flow].

Response 5: Thank you very much for your comment. In this revised version, we have improved the logical flow by using clearer sentences and transition words (lines 58-74).

Comments 6: [Lines 118–134: Condense background justification into more concise statements directly linked to the study objectives].

Response 6: Thank you very much for your comment. We have written a condensed and more direct version of the paragraph, focusing the background justification on the specific objectives of our study (lines 115- 128).

Comments 7: [Lines 138–148: Rephrase sentences to reduce redundancy around sample recruitment].

Response 7: Thank you very much for your comment: We wrote a revised and condensed version of the paragraph on sampling and recruitment, removing redundancy and improving flow (lines 140- 155).

Comments 8: [Lines 163–212: Streamline the description of the S-EDE-Q instrument; focus on its relevance, subscales, and cut-off scores rather than repeating definition].

Response 8: Thank you very much for your comment. Here we provide a simplified description of the S-EDE-Q instrument, focusing on its relevance, structure, and cut-off scores, and eliminating redundant definitions and methodological details. (Lines 167- 181).

Comments 9: [Lines 213–239: Anthropometric procedures could be shortened by removing step-by-step measurement details not essential for replication.]

Response 9: Thank you very much for your comment. Here we provide a simplified description of the anthropometric procedures, focusing on the essential information for replication and eliminating redundant details about step-by-step physical positioning. (Lines 193- 200).

Comments 10: [Lines 260–296: Specify statistical methods more succinctly; clearly indicate which tests were used for each variable.]

Response 10: Thank you very much for your comment. Here we provide a simplified and structured description of our statistical methods, clearly indicating the tests used for descriptive, correlational, and between-group analyses. (Lines 227- 253).

Comments 11: [Ensure consistent reporting format for p-values (e.g., 'p < 0.05', 'p = 0.031')]

Response 11: Thank you very much for your observation. We have been consistent in p <0.05 in most of the results. Specifically, it has been corrected in the lines: 279, 319, 320, 328, 330, 340, 341, 345, 366, 367)

Comments 12: [Lines 309–341: Simplify the description of anthropometric and socio-demographic characteristics. Remove narrative repetition of table content].

Response 12: Thank you very much for your observation. Here is a streamlined summary of the socio-demographic and sport-related characteristics, focusing on key findings and eliminating redundant table references and definitions (Lines 273- 292).

Comments 13: [Table 2 and Table 3: Standardize formatting of percentages and decimal points].

Response 13: Sorry, but we believe that both the percentages and decimal points in Tables 2 and 3 are in the same format, unless we are mistaken. However, before final editing, the format of all tables will be reviewed to ensure consistency.

Comments 14: [Lines 357–397: Ensure clarity in figure captions and statistical terms (e.g., 'AME' should be defined upon first use)].

Response 14: Thank you very much for your observation. The captions for figures 3 and 4 have been clarified.

Comments 15: [Figures 3–5: Consider simplifying figure legends to highlight main findings only.]

Response 15: The captions for Figures 3-5 have been simplified.

Comments 16: [Lines 497–534: Improve alignment between the main findings and the literature cited; avoid overgeneralization].

Response 16: Thank you very much for your suggestion. The revised manuscript presents the discussion  in which the presentation of the results is simplified and the link with the bibliography is reinforced (Lines 455- 481)

Comments 17: [Lines 539–577: Some sentences are excessively long and could be split for clarity]

Response 17: Thank you very much for your suggestion. We have rewritten this part of the discussion with clearer, more concise sentences and logical paragraphs, focusing on the key findings and their alignment with the literatura (Lines 482- 512).

Comments 18: [Lines 579–614: Clarify the interpretation of competition level and position-related findings; ensure statements are supported by the data].

Response 18: Thank you very much for your suggestion. We have rewritten this part of the discussion, ensuring the interpretation of the results related to the level of competence and position, which guarantees a strong alignment between our data and the literatura. (Lines 513- 537).

Comments 19: [Lines 648–666: Strengthen recommendations regarding early detection strategies and the role of coaches].

Response 19: Thank you very much for your recommendation. In this paragraph, we have revised our observations, reinforcing the recommendations regarding early detection and the crucial role of coaches, using direct and impactful language. (Lines 583-617).

Comments 20: [Improve coherence between the descriptive and analytical findings and the conclusions drawn].

Response 20: We believe that this observation can be answered in the following comments.

Comments 21: [Lines 713–720: Rephrase for conciseness and impact; clearly highlight key implications for practice and future research].

Response 21: In this manuscript, we attempt to answer this interesting question, clearly highlighting the key implications for both sports practice and future research. (Lines 658- 671).

Comments 22: [Emphasize the exploratory nature of the study and the need for larger longitudinal research].

Response 22: We have reformulated the conclusions, emphasizing the exploratory nature and the critical need for large-scale longitudinal research to confirm the preliminary results.

Comments 23: [Ensure uniform formatting of references throughout (capitalization, punctuation, DOI formatting)].

Response 23: We have revised the references as suggested by the reviewer

Comments 24: [Verify consistency in citation numbering and in-text referencing].

Response 24: We have revised the numbering of citations and references in the text.

Comments 25: [Remove redundant or repeated references if present].

Response 25: Dear reviewer: from our point of view, we have not found any duplicate or redundant references.

Round 2

Reviewer 1 Report

Comments and Suggestions for Authors

None

Reviewer 2 Report

Comments and Suggestions for Authors

Dear Authors,

After review, the authors responded to all questions, comments, and revisions, so I consider the article acceptable in its current form.

Congratulations.